# Peronosporales Species Associated with Strawberry Crown Rot in the Czech Republic

**DOI:** 10.3390/jof8040346

**Published:** 2022-03-26

**Authors:** Matěj Pánek, Marie Maňasová, Jana Wenzlová, Miloslav Zouhar, Jana Mazáková

**Affiliations:** 1Team of Ecology and Diagnostics of Fungal Plant Pathogens, Crop Research Institute, Drnovská 507/73, 161 06 Prague, Czech Republic; 2Department of Plant Protection, Faculty of Agrobiology, Food and Natural Resources, Czech University of Life Sciences Prague, Kamýcká 129, 165 00 Prague, Czech Republic; manasova@af.czu.cz (M.M.); wenzlova@af.czu.cz (J.W.); zouhar@af.czu.cz (M.Z.); mazakova@af.czu.cz (J.M.)

**Keywords:** oomycetes, *Phytophthora cactorum*, root pathogens, strawberry disease, root rot

## Abstract

The symptoms of crown rot on strawberry plants are considered typical for the pathogen *Phytophthora cactorum*, which causes high losses of this crop. However, an unknown number of related species of pathogens of Peronosporales cause symptoms quite similar to those caused by *P. cactorum*. To determine their spectrum and importance, strawberry plants were sampled from 41 farms in the Czech Republic. The cultures were isolated from the symptomatic plants using the baiting method, with subsequent cultivation on a semiselective medium. Isolates were identified to the species level using nuclear ribosomal internal transcribed spacer (ITS) barcoding after preliminary morphological determination. In total, 175 isolates of 24 species of *Phytophthora*, *Phytopythium*, *Pythium,* and *Globisporangium* were detected. The most represented was *Phytophthora cactorum,* with 113 (65%) isolates, which was recorded in 61% of farms, and the *Pythium dissotocum* complex with 20 (11%) isolates, which was recorded in 27% of farms. Other species were represented in units of percent. Large differences between farms in the species spectra were ascertained. The differences between species in cardinal growth temperatures and different management of the farms are discussed as a main reason for such a diversification. Regarding the dissimilar sensitivity of various species of Peronosporales against fungicides, the proper determination of the cause of disease is of crucial significance in plant protection.

## 1. Introduction

Strawberries, which are grown in convenient environments of temperate climate zones worldwide, are important crops. The most devastating pathogens of this plant species include fungi such as *Botrytis cinerea* and *Verticilium dahliae* [1,2], but one of the most serious pathogens is an oomycete of the order Peronosporales—*Phytophthora cactorum* (Lebert and Cohn) J. Schröt. This polyphagous pathogen causes damage to strawberry plants by attacking the roots and rhizomes and inflicting extensive necroses [3]; under some circumstances, it also infects fruits [4,5]. Under the weather conditions convenient for disease development, losses can reach up to tens of percent [6]. The typical symptoms of infection are the wilting of leaves and the creation of reddish to brownish lesions often visible on the cross-section of the rhizome. The symptoms on underground plant organs are necroses on gradually blackening roots, which simultaneously lose their ability to supply the plant with water and nutrition [4]. Under conditions beneficial for pathogen development, the infected plants often die in a few days. On the affected fruits, *P. cactorum* causes leather rot and softening of the fruit. The ability of *P. cactorum* to spread via actively movable flagellated zoospores is immense under circumstances convenient for spreading, i.e., in water or water-saturated soil [7,8]. Additionally, the ability to survive viable in soil via thick-walled oospores is significant [9,10,11]. Although the detrimental influence of this pathogen on the health of strawberry plants has been known for a long time [4], considerably less is known about the importance of related species of soil-borne pathogens of the same order, Peronosporales, which are characterized by similar abilities [8]. As the exact causes of diseases caused by various species of these pathogens are difficult to distinguish from each other based only on symptoms, part of the damage caused by those pathogens is probably incorrectly attributed to *P. cactorum* [8].

During the last few decades, in different parts of the world, many pathogenic species of the order Peronosporales have been documented in association with strawberry plants. Dissimilar to *P. cactorum*, none of them seemed to have a similar regular association with strawberries. Nine species have been documented from the genus *Phytophthora,* which are associated with strawberries, including some of the most important polyphagous pathogens, such as *P. fragariae*, *P. plurivora*, *P. citophthora*, *P. nicotianae*, *P. cryptogea*, *P. bisheria*, *P. nagaii*, *P. fragariaefolia,* and *P. capsici* [12,13,14,15,16,17,18,19]. Associations with strawberries have been documented in other soil pathogens of the related genera *Globisporangium*, *Elongisporangium*, *Pythium*, and *Phytopythium*. Some of their members were only recently separated from the former genus designation *Pythium* [20,21]. As a cause of strawberry disease, the following species were identified: *Globisporangium ultimum* (all *Globisporangium* spp. in association with strawberry plants were originally documented as *Pythium* spp.), *G. intermedium*, *G. violae*, several members of the *G. sylvaticum* complex, *G. spinosum*, *G. echinulatum*, *G. paroecandrum*, *G. carolinianum*, *G. spinosum*, *G. intermedium*, *G. mastophorum* and *G. polymastum*, *Elongisporangium* (as *Pythium*) *anandrum*, *Pythium afertile*, *Py. angustatum*, *Py. apleroticum*, *Py. inflatum*, *Py. torulosum*, *Py. aphanidermatum*, *Py. myriotylum*, *Phytopythium helicoides*, and *Phy. megacarpum* [20,22,23,24,25,26,27,28,29]. Some of these species, together with several members of the fungal genera *Fusarium* and *Rhizoctonia* as well as nematodes, such as *Pratylenchus penetrans*, are believed to participate in strawberry black rot disease, which is a complex disease of strawberry plants [30,31,32,33].

Since the ability of many of these *P. cactorum* relatives to survive and spread is similar to that of *P. cactorum* alone [8,34,35,36], it is reasonable to evaluate the importance of these pathogens for strawberry disease development. The aim of the current study is to evaluate the presence of soil-borne pathogens of Peronosporales in association with strawberry plants showing symptoms resembling disease caused by *P. cactorum* in farms in the Czech Republic. The study is based on the isolation of pathogen strains from plant roots using the leaf baiting method with subsequent cultivation on semiselective media. Species determination was performed by the commonly used method, nuclear ribosomal internal transcribed spacer (ITS) DNA barcoding [36,37,38,39,40], after preliminary morphological classification.

## 2. Materials and Methods

### 2.1. Sampling

A total of 41 strawberry farms in the Czech Republic was sampled, covering the whole spectrum from small farms comprising only a few fields to the largest ones, focused on intensive commercial strawberry production. Strawberry plants showing typical symptoms of *Phytophthora* crown rot (wilting, greyish leaves, necrotic roots, or reddish-brown lesions on cross sections of rhizomes) [4] were found on the farms, and samples of plants showing such symptoms were taken. The pathogens were isolated using the leaf baiting method [41]. The roots and rhizomes of symptomatic strawberry plants were washed with tap water and flooded with demineralised water in a plastic dish. The baiting leaves of susceptible plant species (*Fagus silvatica*, *Hedera helix*, *Castanea sativa*, *Rhododendron* sp.) were placed on the water’s surface. After the necrotic lesions developed on these leaves, the whole leaves were washed with tap water and surface sterilized with 70% ethanol for 30 s. The border regions between the affected and healthy tissues on the margins of lesions were cut out and placed on the semiselective medium PARPNH V8 [42,43]. The cultures developed from the specimens of plant tissues were then maintained on V8 agar plates. The isolates were deposited in the culture collection of agriculturally important fungi of the Crop Research Institute, VURV-F.

### 2.2. Species Determination

After a morphological determination based on the morphology of reproductive structures present (oogonia, zoosporangia, chlamydospores), all cultures were classified into species using ITS DNA barcoding [44]. For molecular processing, the cultures were cultivated on V8 agar plates covered by cellophane wrap. After the colony was developed, the mycelium was scraped from the surface of the cellophane, and a small piece was placed into a microcentrifuge tube in 20 µL of a dilution buffer of the Phire Plant Direct PCR Kit (Thermo Fisher Scientific, Inc., Waltham, MA, USA). This kit was also used to perform the PCR according to the manufacturer’s instructions; 2 µL of dilution buffer containing the sample DNA was added to the total volume of 25 µL of Phire Plant PCR Master Mix, including 0.2 mM primers ITS1 and ITS4 [45]. The reaction was performed in an Eppendorf Mastercycler Nexus Thermal Cycler (Eppendorf AG, Hamburg, Germany) under the following settings: 98 °C for 2 min; 35 cycles of 98 °C for 30 s; 55 °C for 30 s; 72 °C for 60 s; then, 72 °C for 10 min. The purification and sequencing of the PCR product were performed by Macrogen, Inc. (Seoul, Korea). The DNA sequences were determined both in the forward and reverse directions; the consensual sequences were used in subsequent analyses. The identity of the species was determined by comparing the DNA sequences to the NCBI database using the BLAST algorithm. In the final species determination, the preliminary ascertained morphological characteristics were also taken into consideration.

### 2.3. Evaluation of Soil-Inhabiting Species Spectra of Peronosporales Present in Particular Localities

During sampling in the field, the number of samples of symptomatic plants taken from each farm, the number of isolates, and the number of species from each sample were recorded. Only one member of each species isolated from one plant was considered an individual isolate. The number of isolates of each species found was expressed as a percentage of the total number of isolates from each farm and from all farms. The number of farms in which no isolate/only *P. cactorum*/*P. cactorum* accompanied by other species/only other species was found were expressed in absolute values and percentages. The correlation between the number of samples, isolates, and species found on each farm was ascertained using the software Statistica 14.0 (Tibco Software Inc., Palo Alto, CA, USA). To express the degree of correlation of these numbers, the Spearman coefficient was calculated, and the same numbers were displayed on the line chart.

## 3. Results

The number of samples taken from particular farms was 1 to 94 depending on the size of the farm and on the number of symptomatic plants found; the average number of sampled plants per farm was 16. In total, 459 strawberry plants were taken showing typical symptoms of infection by *Phytophthora cactorum*. After the isolation of mycelial cultures from plant specimens, 24 species were identified (Table 1). The detailed list of all isolates and the NCBI GenBank accession numbers of sequences of their ITS rDNA region used in species determination are given in Appendix A. Since the unambiguous identification of some species closely related to *Pythium dissotocum* was not possible on the basis of either sequencing or morphology [46], these isolates were labelled the *Pythium dissotocum* complex, which was in agreement with the solution used in other works [36,47].

Considering only one member of each species isolated from one plant as an individual isolate, a total of 175 cultures was isolated. In only six cases, two different pathogen species were isolated from one plant. The majority (113, i.e., 65%) of isolates were identified as *Phytophthora cactorum*; other frequently represented isolates included members of a *Pythium dissotocum* complex (20, i.e., 11%). Apart from those two species, the other species were represented by less than ten percent (Table 1). The proportion of farms where only *P. cactorum* was sampled was 32%; in another 29% of farms, this species was accompanied by other pathogens of the genera *Pythium*, *Phytopythium*, or *Globisporangium*. In 22% of farms, only pathogens other than *P. cactorum* were revealed; in seven farms (17%), no soil-borne pathogens of Peronosporales were isolated (Table 2). Regardless of the species to which they belong, the number of isolates sampled in farms was mostly one to four, with an average of four per farm. The maximum number of isolates originating from one farm was 24; this number represented three species. The maximum number of species originating from one farm was nine, although such a high number was found in only one farm (2%). Only one species was found in twenty farms (49%), while two species were found in nine farms (24%) (Table 3, Appendix A). These results not only demonstrate that *P. cactorum* substantially predominates over other soil-borne Peronosporales pathogens, but also indicate the significant participation of at least another 1–3 species of this family in plant losses.

A correlation was found between the number of isolates and the number of species originating from the same farm (Spearman r = 0.74, *p* < 0.05, Table 4, Figure 1). The correlation between the number of sampled plants on each farm and the number of isolates was lower (r = 0.48, *p* < 0.05); the correlation between the number of sampled plants and the number of species found was even lower (r = 0.19, *p* < 0.05).

## 4. Discussion

The species spectra recorded in particular farms substantially differed from each other, although the distances between farms sampled were lower than a few hundred kilometres. Regardless of the incomparably lower sensitivity of the method we used, i.e., baiting/cultivation, in comparison to the currently widely used metabarcoding, this method had the advantage of capturing only substantially vigorous organisms [48,49]. However, this limitation was common for all sampled farms. In contrast, the metabarcoding method has the disadvantage of a lower ability to clearly differentiate between some species, but most importantly, this method does not have the ability to differentiate between dead and living microorganisms [48]. As follows from the correlation analysis (Table 4, Figure 1), the number of species we found on farms only indirectly and loosely depended on the sampling intensity. Except for the frequency of the occurrence of symptomatic plants, the sampling intensity roughly reflected the size of the farm. Therefore, in the current study, the approximate capture of the true spectrum of oomycete pathogen species present in the rhizosphere of strawberry plants could be assumed, and the total number of species found probably would not have increased substantially even if more plants had been sampled.

In the study, 24 species belonging to four genera of Peronosporales were isolated from symptomatic strawberry plants. All of these species have a global distribution, and all of them have a relatively wide host spectrum; the majority also included strawberries, although in some species, the host spectrum has not yet been well explored in detail [29,30,31,47,50,51,52,53,54,55,56,57,58,59,60,61,62,63,64,65,66,67,68,69,70,71,72,73,74,75,76,77].

The most important strawberry pathogen we found was *Phytophthora cactorum*, which was found in 61% of the farms. Although this species dominated in the majority of farms we sampled, in 22% of farms, we did not find this species, while other soil-borne pathogens of Peronosporales (*Phytophthora pluvivora, P. cryptogea, Pythium dissotocum* complex, *Py. nodosum, Py. ultimum, Py. aphanidermatum, Py. heterothallicum, Py. salpingophorum, and Phytopythium montanum*) were successfully isolated. In an additional 15% of farms, we did not find any of these pathogens, although unsuccessful baiting cannot be simply interpreted as evidence of their absence. Regardless, it seems that *P. cactorum* could be missing for a significant proportion of farms, where the symptoms on plants could have been caused by other related members of Peronosporales. Thus, the distribution of *P. cactorum* did not meet the basic assumption of a standard distribution model, i.e., the equilibrium of the distribution of indigenous organisms with the environmental conditions [78]. Since this is a typical feature of invasive species, such an irregular distribution supports the conception of the non-European origin of this pathogen. The origin of the entire *Phytophthora* Clade 1, including *P. cactorum*, is assumed to be North America [79], although it has not yet been unambiguously evidenced. The relatively short coevolution time between strawberry plants and *P. cactorum* has been documented by the exclusive presence of only one of five known genetic lineages of this species on strawberry plants in the Czech Republic [80], although other lineages are also present on woody host species in this region [81]. Such an exclusive presence of one lineage on strawberries has already been evidenced and explained in association with the intraspecific host specificity developed in *P. cactorum* [82,83,84].

Three members of Clade 6 [85] of the genus *Phytophthora* were found in strawberry farms; *P. bilorbang*, *P. lacustris,* and *P. cryptogea*. The association of *P. lacustris* with strawberry plants was documented only recently [86], while to our knowledge, the association of *P. bilorbang* with strawberries has not yet been recorded. However, since this species was originally described as a pathogen of *Rubus anglocandicans* (Rosaceae) [87], its association with the strawberry, a member of the same plant family, is not surprising. Both of these two pathogen species are associated with wetlands or periodically flooded localities. They are the most frequently found in association with (semi)natural stands of riparian forests throughout Europe [50,51,52,53,54]. The possible explanation of their occurrence in strawberry farms could, thus, be hypothesized as a consequence of irrigation with unfiltered water from watercourses or ponds contaminated by these *Phytophthora* species, by flooding of fields by contaminated water in association with some climatic extremes, or by any other accidental event. In contrast to these two previous species, *P. cryptogea*, which has also been mentioned to occasionally be associated with strawberry plants [88,89], has an evidenced frequent occurrence in agriculture, although it occurs in forestry and nurseries as well [55,56]. In Europe, this species is common on crops and ornamental plant species, as well as in varied natural stands of woody plants [79,90]. Strains of *P. cryptogea* differ from each other in their host range [91], i.e., host specificity is evolved in this species. Some isolates are also pathogenic to wheat and other crops, which is congruent with their presence in agricultural lands [92].

The members of *Phytophthora* Clade 6 are considered usually saprophytic or only occasionally pathogenic in their original forest habitats under normal conditions [93], although the importance of some of them may be underestimated, and some of them are also able to cause significant plant damage [94]. For some Clade 6 *Phytophthora* spp. such as *P. lacustris* and *P. bilorbang,* the development of the infection is also hypothesized to be highly influenced by the temperature. These species switch to a pathogenic lifestyle from that saprophytic lifestyle only under highly optimal environmental conditions [94].

In contrast to the previous species, *P. plurivora* is considered one of the most important *Phytophthora* pathogens worldwide, which is associated mainly with forests [79]. This species was suggested by Schoebel et al. [95] to be indigenous to Europe, but given its asymptomatic occurrence on trees in local undisturbed native forests as an indirect indication of long-term coevolution [96], the origin of *P. plurivora* and all of *Phytophthora* Clade 2 seems more likely to be in Southeast Asia [97,98]. Nevertheless, *P. plurivora* is one of the most frequently sampled *Phytophthora* species in European forests in diverse natural and seminatural stands, as well as in ornamental and horticultural nurseries [79], while its occurrence in association with agriculture and strawberry farms is only rare [18,99]. The other member of Clade 2 we sampled, *P. citrophthora,* has a main distribution area in the Mediterranean basin and similar regions in association with various *Citrus* species [100]. Although *P. citrophthora* causes the greatest damage to *Citrus* spp., it also has a wide spectrum of other hosts, including the strawberry [55,57]. However, even this pathogen has not yet been shown to be a regular threat to this crop.

Although the origin of the majority of the mentioned *Phytophthora* species remains unknown, the origin of all other members of the order Peronosporales that we isolated is even less clear. Based on the number of strains we isolated, the second most important pathogen after *P. cactorum* was the *Py. dissotocum* complex. Comparing the results of the current study to the other one performed by Toljamo et al. [27] in strawberry fields in Finland, the spectrum of species of the order Peronosporales was shown to be completely different from the spectra found in the current study, where only *P. cactorum* and *Py. dissotocum* were isolated equally. *Pythium dissotocum* and other related pathogens were recently shown to be able to cause necrosis on strawberry plants of similar size as *P. cactorum* [101]. Since *Py. dissotocum* was isolated from symptomatic plants in 11 farms (i.e., more than one-quarter of them), and more than 10% of all isolates belong to this species, this should not be considered an unimportant pathogen of strawberry plants. This species, together with *Py. aphanidermatum*, belongs to the most serious pathogens of this genus for crop production. Both species are associated mainly with vegetables [102]; however, the association with strawberry plants has also been documented [58,59]. We isolated two other members of this genus, *Py. intermedium* and *Py. mamillatum,* which are already known pathogens of strawberries [59,103]. The total host spectrum of the majority of other *Pythium* spp. we isolated (*Py. rotratifingens*, *Py. montanum*, *Py. conidiophorum*, *Py. nodosum*, *Py. perplexum*, *Py. salpingophorum*, *Py. Nodosum,* and *Py. torulosum*) remains unknown, and the pathogenicity of some of them against particular host species is not clear. This is obvious in the example of *Py. perplexum*, which is documented as both pathogenic and nonpathogenic on beans [104,105]. To our knowledge, this was the first report of all eight *Pythium* species being associated with strawberry plants. Since we did not perform an exact test of their pathogenic role on this crop, theoretically, their role in strawberry roots could be saprotrophic. However, their phytopathogenic nature is well known in many other host species, and since they were isolated from strawberry plant tissues, their role in damage to this host is quite probable.

The differences in species spectra recorded in the particular farms we investigated (Appendix A), as well as the noticeable differences between the entire species spectrum found in the current study and in that of Toljamo et al. [27] in Finland, can be explained as a result of a combination of two phenomena. Mainly during the biotrophic phase of infection, the formation and severity of plant disease greatly depend on the temperature [102,106]. This phenomenon has already been mentioned by Littrell et al. [107] for *Pythium aphanidermatum*, by Watanabe et al. [29] for *Globisporangium ultimum*, and by Grove et al. [108] for *Phytophthora cactorum*. This optimal temperature for infection can be quite different from the optimal temperature for growth [29]. The growth temperatures of the species of Peronosporales we found differed considerably, which was documented by the range between the lowest and highest optimum temperatures of these species as wide as 20–35 °C [51,59,61,63,68,72,87,97,103,109]. Although little is known about the temperatures beneficial for infection, the optimal temperature for the development of infection by *Globisporangium ultimum* is between 9 and 20 °C [29], while that of *Pythium aphanidermatum* is between 30 and 35 °C [59]. The total range of the cardinal temperatures of a whole species spectrum we isolated was even wider. The composition of the entire community of these pathogens very likely reflects such differences, which pose a threat considering potential climate changes, because the increased temperatures have the potential to cause alterations in the local spectra of pathogens. Such a change could also cause some of the originally rather saprotrophic Peronosporales to switch to a pathogenic lifestyle, as was described for the case of *Phytophthora* spp. of Clade 6 [94]—*P. lacustris* and *P. bilorbang*, the species that we also found. In addition, these two species are also considered tolerant to higher temperatures [51,87], which is potentially beneficial for them in changed environmental conditions.

Another phenomenon probably participating in differences between the species spectra we identified lies in the long-term accumulation of pathogenic species in soil in cases where one crop is cultivated in the same field for a long time. This phenomenon was described for *Pythium* species in maize and soya fields [110,111] and for many plant pathogens, including some Peronosporales in strawberries [103]. Our observations appeared to be in accordance with these conclusions, as fields used for intensive and long-term commercial strawberry production on farms using pregrown plant stocks are the most severely contaminated with soil Peronosporales pathogens (data not shown), while small fields are less affected.

Both of the mentioned explanations of the presence of different species spectra, whose combination is a probable principal cause of the presence of such a wide total species spectrum and of differences between local spectra, constitute a certain danger for strawberry production. Except for temperatures favouring some non-native species, the joint presence of allopatric species in one field poses a risk of the hybridization of such species [112], with accompanying effects, such as changes in the host species range, increased invasiveness, changes in the ecological requirements of the new hybrid, or the transmission of the genes of resistance to fungicides into a new context. The instability of newly created genomes [113], also associated with potential polyploidy or aneuploidy, enables the increased development of adaptations [114], improved fitness, and the creation of new properties of hybrid progenies [115,116], although the detrimental influence on the progeny, such as the increased oospore abortion rate of some of the descendants or their decreased fitness, could also be the result of hybridization [113]. Such a development can be particularly increased if the progenies further reproduce asexually or substantially asexually [117,118]. The environmental changes, together with the quite different temperature demands of participating parental species, present favourable conditions for such development. The small, genetically plastic populations [119] potentially represented by only a few newly arisen genotypes are able to readily respond to the changing environment.

Most of the *Phytophthora* species we found have not yet been shown to be systematically associated with strawberry plants, and in the current study, they were recorded only sporadically. Thus, the ascertained occurrence of *P. plurivora* and *P. citrophthora* on strawberry plants seems to be more likely the result of an accidental infection from some surrounding sources than the systemic infection spread, for example, by planting stock. However, similar to *P. cactorum*, species such as *P. cryptogea* and *P. plurivora* [84,91,120,121] and probably *P. citrophthora* as well [122,123] have evolved host specificity at the intraspecific level, meaning that particular pathogen species strains have an uneven ability to infect diverse host species, with one lineage having an increased ability to infect one host species. Due to their immense capacity for clonal reproduction, after accidental contact with strawberry plants, the particular strain of these *Phytophthora* species or some new hybrids coincidentally better adapted to this host could give rise to a strawberry-specific lineage with increased pathogenic potential against this crop. The association of these species with strawberry plants has also been documented in other works [18,55,57], and the spread of *Phytophthora* species from nurseries to plantings has also been described [79]. The ability of some members of the genera *Phytophthora*, *Pythium*, *Phytopythium,* and *Globisporangium* to cause strawberry plant damage comparable to that caused by *P. cactorum* has been demonstrated [101], and these species are also differently sensitive to fungicides [124]. In the case of the adaptation of these species to strawberry plants, due to the common use of pregrown plantings replanted from infested fields, the fast spread of such lineages can be expected.

## 5. Conclusions

The results of our research showed that in addition to *Phytophthora cactorum*, many other related species participate in damage to strawberry plants, which were formerly almost exclusively attributed to *P. cactorum*. These polyphagous species are members of *Phytophthora*, *Pythium*, *Phytopythium,* and *Globisporangium*. Some of these species are known as important pathogens associated with some crops, including strawberries, or more generally with agriculture. The other species we also found have been considered to be associated with natural and seminatural habitats. Their presence in fields poses risks in terms of their potential change from the mainly saprotrophic to a parasitic mode, which could be mediated by climate change. Another risk is the potentially accidental development of their lineages specific to strawberry plants, as well as the potential for the creation of hybrid species through hybridization between formerly allopatric parental species. The different sensitivity to fungicides of various, often nonindigenous, species poses another important threat associated with their presence in strawberry fields.

## Figures and Tables

**Figure 1 jof-08-00346-f001:**
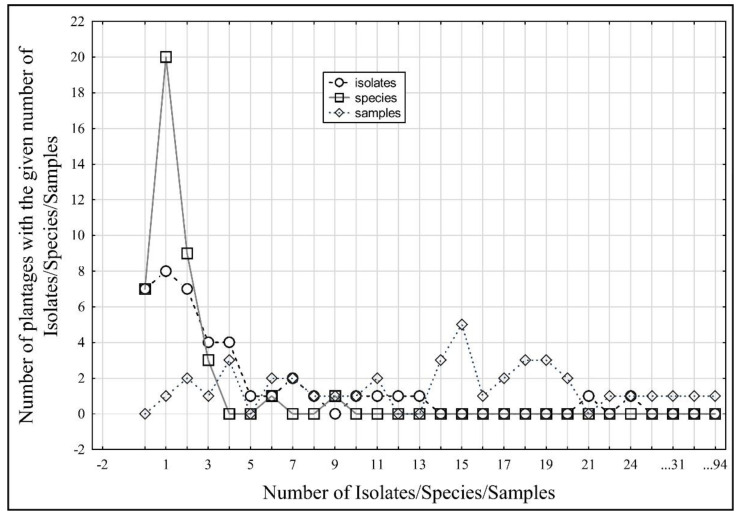
The correlation between numbers of sampled plants, numbers of isolates of Peronosporales and numbers of species of Peronosporales isolated from symptomatic strawberry plants on the farms in the Czech Republic.

**Table 1 jof-08-00346-t001:** List of the species of the order Peronosporales isolated in strawberry fields in the Czech Republic.

Species Name	No. of Isolates Obtained	% of Particular Species on the Total No. of Isolates
*Globisporangium irregulare*	3	1.71
*Globisporangium ultimum*	3	1.71
*Phytophthora bilorbang*	1	0.57
*Phytophthora cactorum*	113	64.57
*Phytophthora citrophthora*	2	1.14
*Phytophthora cryptogea*	1	0.57
*Phytophthora lacustris*	1	0.57
*Phytophthora plurivora*	1	0.57
*Phytopythium citrinum*	1	0.57
*Phytopythium mercuriale*	1	0.57
*Phytopythium montanum*	1	0.57
*Phytopythium vexans*	5	2.86
*Phytopythium litorale*	1	0.57
*Pythium aphanidermatum*	1	0.57
*Pythium conidiophorum*	1	0.57
*Pythium dissotocum* complex	20	11.43
*Pythium heterothallicum*	3	1.71
*Pythium intermedium*	4	2.29
*Pythium mamillatum*	2	1.14
*Pythium nodosum*	5	2.86
*Pythium perplexum*	1	0.57
*Pythium rostratifingens*	1	0.57
*Pythium salpingophorum*	1	0.57
*Pythium torulosum*	2	1.14

**Table 2 jof-08-00346-t002:** The numbers and percentages of strawberry farms in the Czech Republic in which *P. cactorum* and other species of the order Peronosporales were recorded.

Presence of Pythiaceae Pathogens	No. of Localities	% of Localities
Not any species found	7	17.1
Only *P. cactorum* found	13	31.7
Only other species than *P. cactorum* found	9	22.0
*P. cactorum* and other species found	12	29.3

**Table 3 jof-08-00346-t003:** Numbers of plant samples, isolates, and species recorded in strawberry farms in the Czech Republic.

No.	No. of Plantations with the Given Number of Sampled Plants	No. of Plantations with the Given Number of Isolates	No. of Plantations with the Given Number of Species
0	0	7	7
1	1	8	20
2	2	7	9
3	1	4	3
4	3	4	0
5	0	1	0
6	2	1	1
7	2	2	0
8	1	1	0
9	1	0	1
10	1	1	0
11	2	1	0
12	0	1	0
13	0	1	0
14	3	0	0
15	5	0	0
16	1	0	0
17	2	0	0
18	3	0	0
19	3	0	0
20	2	0	0
21	0	1	0
22	1	0	0
23	0	0	0
24	1	1	0

**Table 4 jof-08-00346-t004:** Spearman correlation coefficient r expressing the correlation between the numbers of isolates, species, and sampled plants in strawberry farms.

	No. of Isolates	No. of Species Identified	No. of Plants Sampled
No. of isolates	1.000000	0.742487	0.485586
No. of species identified	0.742487	1.000000	0.191530
No. of plants sampled	0.485586	0.191530	1.000000

## Data Availability

The data presented in this study are available on request from the corresponding author.

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
