# Peer review of "Peronosporales Species Associated with Strawberry Crown Rot in the Czech Republic"

_jof, 2022, doi:10.3390/jof8040346_

Round 1

Reviewer 1 Report

Pánek et al. collected strawberry samples showing Phytophthora crown rot from 41 farms in the Czech Republic and isolated oomycetes of the order Peronosporales, including one of the most severe pathogens of strawberry, Phytophthora cactorum. The authors aimed to investigate the importance of oomycete species in disease development on strawberry. The authors found that the number of species belonging to the order Peronosporales was not closely related to the number of samples. The authors used the baiting method to isolate oomycetes of the order Peronosporales. As the authors pointed out, this method has a disadvantage compared to metabarcoding; however, they chose the suitable method with the advantage of capturing specifically oomycetes. In the future, the pathogenicity test has to be performed to confirm whether several species of Pythium and Phytopythium, whose pathogenicity on strawberry have been unknown, are pathogenic to strawberry. Also, the pathogenicity test of a combination of species found in the same farm would be valuable to find the way to control Phytophthora disease on strawberry. This manuscript is well-organized and well-written. I have only a few minor concerns.

Minor concerns

L85 Phytophthora crown rot

L99 Please remove ‘a rough'

L117 What is 'soil species'? Is ‘soil’ a typo?

L146 identified belongs to the species → were identified as

L159 Is not the number of farms where two species were found nine, based on the data in table 3? Which one is correct, the number in the text or table 3?

L289-290 As the authors mentioned, the pathogenicity of eight Pythium species has not been examined on strawberry plants and some of them are even not known to be pathogenic. Wouldn’t they be saprophytes? I think the authors could declare that this is the first report of eight Pythium pathogens only if the pathogenicity of these isolates was confirmed on strawberry plants. I would suggest the authors rewrite the sentence.

Author Response

Dear reviewer,

On behalf of whole authors collective I would like to express our thanks for your recommendations, all of which we thoroughly considered and used to improve the quality of whole manuscript. We edited the text in all indicated places in the way you recommended.

The best regards

MatÄ›j Pánek, corresponding author

Reviewer 2 Report

Dear Authors,

There are several important points that are missing in your paper

  1. The abstract should be re-written as it missing objectives of the study
  2. English must be improved.  
  3. For several sections, citations are missing, as the whole paragraph has one citation.
  4. In the introduction, I do not think it is necessary to add the diseases cycle of Phytophthora cactorum.
  5. you need to emphasise why these fungi and their disease are important in your regions and thus why this study is important.
  6. when providing numbers, consider rounding them off, isn't it odd to have 0.8% farms, it is ok as 1%
  7. species confirmation is the key important factor in any mycology research. where is your data on how you identify these isolated species? if use only ITS, did you BLAST them? Run a phylogenetic tree?    in addition to that, all comments are given in PDF.

Author Response

Dear reviewer,

On behalf of whole authors collective I would like to express our thanks for your recommendations, all of which we thoroughly considered and used to improve the quality of whole manuscript. We modified the text according to recommendations you sended in .pdf file. The responses to your suggestions and questions I placed directly next to them in the text bellow.

The best regards

MatÄ›j Pánek, corresponding author

1. The abstract should be re-written as it missing objectives of the study

Response: The Abstract was modified, completed by reasoning of study and by some conclusions indication.

2. English must be improved.  

Response: Whole manuscript was english proved by native english speaker already before the first submission, and again after the editing on reviewers recomendations.

3. For several sections, citations are missing, as the whole paragraph has one citation.

Response: Originally we did not place references in some places of introduction part when we supposed the information mentioned as general or generally known, or the sense of the information did not focus on the main issue of the paper. Recently we added few references there to improve the intellibility of text and believe, the total number of references is high enough.

4. In the introduction, I do not think it is necessary to add the diseases cycle of Phytophthora cactorum.

Response: This part has been intended as the explanation highlighting the features of questioned pathogens common for all of them, i.e., i.e., high ability to spread and to durate viable in soil. On the request we shortened this part significantly.

5. you need to emphasise why these fungi and their disease are important in your regions and thus why this study is important.

Response: We believe, the mention that P. cactorum is classified between the most important pathogens of strawberry plants epmphasis the importance of the study enough and the possibility of mistaken identification of cause of strawberry plants disease clearly explain another associated risks. But we added new sentence (line 39 – 40) emphasising the importance of this issue just more. We also added the note of theoretically similar abilities to P. cactorum in related species (line 63). In addition, whole discussion part describe the importance of some pathogens we have worked with, the importance of our results is expressed in Conclusions and we complete some mention also into Abstract, although we tried to not increase the length of this part overly as well.

6. when providing numbers, consider rounding them off, isn't it odd to have 0.8% farms, it is ok as 1%

Response: We accpted this suggestions and rounded all numbers in the text. The exact values have been retained in tables.

7. species confirmation is the key important factor in any mycology research. where is your data on how you identify these isolated species? if use only ITS, did you BLAST them? Run a phylogenetic tree?   

Response: All sequences were aligned to NCBI databse records by BLAST algorhitm, as we already described in original of our manuscript (M&M, lines 120 – 121).

The all sequences we used in isolates determination are listed in the Supplementary table S1, which has been mentioned in the text (Lines 141 – 143).

We believe the construction of phylogenetic tree is undoubtly possible, but since the genera we found are recently considered as not the members of common family (doi.org/10.3114/fuse.2019.03.10), such tree does not seem to be very valuable because of missing members of other related genera/families. Phylogeny based on only one DNA locus is not good precondition for valuable tree as well, this should be more probably performed based on some WGS method. Finally, we do not assume there is some direct association between phylogeny and species spectrum of species we found out.

Round 2

Reviewer 2 Report

Dear Authors

I have several comments on your revised version. The species identification section is I am not stratified. You just mentioned that can not use either morphology or molecular to define species. It's not a good answer. I wonder why you do not have any images relevant to those cultures or the materials used in this study. In addition, other comments are given in the file.

Author Response

Dear Reviewer,

On behalf of whole authors team I would like to express many thanks for another very detailed comments and suggestions you sended. We accepted many of your suggestions. However, we can only guess many of your highlighting words were motivated by the language issues, because usualy not any remarks were placed there. I can assure you, before the submission the whole manuscript was proved for correct english by a native english speaker in respected agency focused on such cases. Therefore we suppose the correctness of the language as unquestionable although probably not of the best elegancy in all cases. The certificate of the proof I appended to the end of this document. In the rest of this letter I will focus on the objective issues you suggested to be solved. The issue you mentioned as the main problem of our work is deeply dissected at our comments for the L 145. Here I can only summarize, this is the problem of taxonomy, not of the species identification. In details we responded to all suggestions and questions in attached .pdf file.

With the best regards

MatÄ›j Pánek